# Concentration of Selected Macronutrients and Toxic Elements in the Blood in Relation to Pain Severity and Hydrogen Magnetic Resonance Spectroscopy in People with Osteoarthritis of the Spine

**DOI:** 10.3390/ijerph191811377

**Published:** 2022-09-09

**Authors:** Marta Jakoniuk, Jan Kochanowicz, Agnieszka Lankau, Marianna Wilkiel, Katarzyna Socha

**Affiliations:** 1Department of Invasive Neurology, Medical University of Białystok, M. Skłodowskiej-Curie 24a Street, 15-276 Białystok, Poland; 2Department of Neurology, Medical University of Białystok, M. Skłodowskiej-Curie 24a Street, 15-276 Białystok, Poland; 3Department of Integrated Medical Care, Medical University of Białystok, M. Skłodowskiej-Curie 7A Street, 15-096 Białystok, Poland; 4Department of Bromatology, Medical University of Białystok, Mickiewicza 2D Street, 15-222 Białystok, Poland

**Keywords:** macronutrients, toxic elements, calcium, magnesium, lead, cadmium, mercury, degenerative spine disease, hydrogen magnetic resonance spectroscopy

## Abstract

Macronutrients and toxic elements may play an important role in the pathogenesis of osteoarthritis of the spine. The objective of this study was to evaluate the relationship between the concentrations of Ca, Mg, Pb, Cd and Hg in blood with the results of hydrogen magnetic resonance spectroscopy and the severity of pain. Patients with osteoarthritis of the spine *(n* = 90) and control subjects (*n* = 40) were studied. The concentrations of mineral components in blood were determined by atomic absorption spectrometry (ASA). Spinal pain severity was assessed using the Visual Analog Scale (VAS). Hydrogen magnetic resonance spectroscopy (^1^H-MRS) was used to determine the fat/water ratio in the bodies of L1, L5 and the L4/5 intervertebral disc. The median concentration of Mg in the serum of subjects with spinal degenerative disease was significantly lower (*p* < 0.001) than that in healthy subjects. The median concentration of Cd in the blood of subjects with osteoarthritis of the spine was significantly higher (*p* < 0.05) than that in the control group. Significantly lower (*p* < 0.05) median molar ratios of Ca to Cd and Pb as well as Mg to Pb and Cd were observed among patients with osteoarthritis of the spine. Significant differences (*p* < 0.05) were observed in the value of the fat/water ratio in selected spinal structures, depending on normal or abnormal serum Ca and Mg concentrations. The study showed some abnormal macronutrient concentrations, as well as disturbed ratios of beneficial elements to toxic elements in the blood of people with osteoarthritis of the spine.

## 1. Introduction

Degenerative spine disease is a chronic, progressive and complex condition. The essence of the disease is the degeneration and premature wear of tissues of a cartilaginous nature, such as intervertebral discs and intervertebral joints. The changes are chronically progressive and irreversible and, over time, affect bone tissue. The location of the disease determines the onset of symptoms. The degenerative process can be primary or secondary, as a consequence of past inflammation, congenital defects of the spine or trauma [1]. To date, the unequivocal cause of the condition has not been established. A large number of factors initiating the degenerative process are now known, which makes it possible to state the polyetiological nature of the condition. The most important of these include age, excessive body weight, abnormal diet, hard physical work, congenital and acquired defects of the osteoarticular system and excessive physical activity [2,3]. Mineral disorders in the skeletal system are increasingly being discussed. Interactions occurring between individual elements and heavy metals can lead to a disruption of their narrow optimal concentration range. Consequently, there is an intensification or weakening of their action, toxic effects on the body or the secondary deficiency of other elements [4,5,6].

Calcium (Ca) and magnesium (Mg) are essential macronutrients for the human body, including for the proper functioning of the skeletal system. The literature data indicate frequent deficiencies of these elements in the diet of adults, because there are many factors that reduce the bioavailability of these ingredients from the diet [7,8]. Ca in the bones occurs as Ca phosphate salts (hydroxyapatites), forming the supporting structure of the skeleton [6]. Chronic inadequate Ca supply in adults, especially postmenopausal women, significantly increases the risk of osteoporosis [9]. Mg conditions the processes involved in bone strengthening and remodeling. Of the body’s pool of total Mg, 60% is found as hydroxyapatite in the skeletal system (25% as a readily exchangeable pool) [10,11]. The normal range of values of Ca and Mg concentrations in the serum of adults is: 86.2–102.2 mg/L and 19.44–24.31 mg/L, respectively [12].

The toxic effects of lead (Pb), cadmium (Cd) and mercury (Hg) are well known. These toxic metals are still common environmental pollutants. Most Pb enters the body through the gastrointestinal tract with food and water (40%), and much less enters through the inhalation and transdermal routes [9]. Bones are a reservoir for Pb and accumulate 90–95% of its amount, rendering it metabolically inactive. Pb is deposited by exchange with Ca^2+^ ions through numerous processes and is then transported to the compact bone, which is much less metabolically active [13,14]. Cd is considered one of the most toxic heavy metals [15]. The best-known example of Cd poisoning is Itai-Itai disease. The condition manifests itself as severe back pain, especially in the lumbar region, as well as bone and muscle pain, demineralization of the skeleton and a characteristic duck-like gait [16]. Hg, even in very low concentrations, poses a threat to living organisms [9]. Hg can accumulate in bone and cartilage tissue, which can cause osteoarticular diseases, especially in people with genetic susceptibility to autoimmunity [17]. The maximum permissible concentrations of Cd, Pb and Hg in the blood, at which no toxic effects are observed, are: 2.7 µg/L, 100 µg/L and 7.2 µg/L, respectively [12].

Hydrogen magnetic resonance spectroscopy (^1^H-MRS) can use a chemical shift to detect water and fat components in the bone marrow and quantify the fat fraction, indirectly informing about bone quality. It opens a new path to early diagnosis and the prevention of osteoarthritis of the spine [18]. In turn, the VAS scale is the most frequently used tool for the subjective assessment of pain severity by the patient. The VAS is an easy-to-use instrument and is highly sensitive in detecting treatment effects, and its results can be analyzed by parametric tests [19].

The objective of the study was to evaluate the relationship between the concentrations of selected macronutrients (Ca, Mg) in serum and toxic elements (Pb, Cd, Hg) in blood with the results of hydrogen magnetic resonance spectroscopy and the severity of pain.

## 2. Materials and Methods

### 2.1. Characteristics of the Study Group and Sample Collection

The material for the study was collected at the Neurology Outpatient Clinic of the Non-public Health Care Facility “KENDRON” in Bialystok. The study included a group of 130 adults aged from 21 to 83 years. Patients with degenerative spine disease (*n* = 90) diagnosed by the Magnetic Resonance Imaging (MRI) SIEMENS Magneton Spectra (Germany) and healthy people (*n* = 40) were included for the study. The control group was appropriately matched in terms of sex, age and percentage of smokers and non-smokers relative to the group with degenerative spine disease. Patients taking dietary supplements were excluded from the study. The mean age in the study group was 48.3 ± 12.9, while in the control group, it was 45.8 ± 16. Information on age, sex and smoking status was collected. Patients and control group characteristics are presented in Table 1.

Venous blood was drawn from the subjects using the vacutainer system test tubes containing a clot activator, which are recommended for elemental analysis (Becton Dickinson, Grenoble, France). The samples were centrifuged for 10 min at approximately 2500 rpm (MPW M-Diagnostic, Med. Instruments, Warszawa, Poland). Blood and serum samples were stored at −20 °C. Written informed consent was obtained from all study participants prior to the collection of blood samples. The study protocol was approved by the Ethics Committee of the Medical University of Bialystok (No. R-I-002/97/2017).

### 2.2. Clinical Examinations of Patients

The intensity of pain was assessed by the patients using the Visual Analog Scale (VAS). Hydrogen magnetic resonance spectroscopy (^1^H-MRS) of selected spine structures was performed in the patients. MRS examinations were performed with the SIEMENS Magneton Spectra (Erlangen, Germany) apparatus with a magnetic field strength of 3T at the Kendron NZOZ Outpatient Clinic. A standard protocol was used in the L-S spine MRI using the sequences T2_TSE, T2_TIRM and T1_TSE in the sagittal plane and T2_TSE in the transverse plane. ^1^H-MRS planning was obtained on T2_SPC sequences (TR; 1300 ms; TE; 208 ms, section thickness 0.9 mm; intersection gap 0.2 mm; 56 sagittal sections; FOV [340 × 170]; and matrix size 288 × 192); from this sequence, the frontal and axial planes with the same technical parameters were reconstructed. ^1^H-MRS was performed with a repetition time (TR) = 2000 ms and an echo time (TE) = 50 ms using the single-voxel spectroscopy (SVS) method, where the assessment of metabolic disturbances (spectrum registration) is possible in a small volume tissue, while it was previously strictly defined on the basis of a conventional (morphological) MRI scan. Voxels (VOI, volume of interest) from the stem L1 and L5 were analyzed, in which the lipid content and the hydration level of the intervertebral disc at the L4 and L5 levels were assessed. A 12 × 12 × 12 mm voxel was used to obtain the spectrum from the vertebral body L1 and L5. On the other hand, the spectroscopic spectrum of the L4/L5 intervertebral disc was obtained using a 20 × 5 × 20 mm promissory note. The spectroscopic spectrum was analyzed using the Siemens dedicated SPECTROSCOPY application. Metabolites were obtained in the form of water peaks and lipids. The level of the L4/L5 intervertebral disc steatosis was determined by determining the lipid-to-water ratio; the ratio level was set at 1 for lipids. The level of steatosis of L1 and L5 vertebrae was determined by determining the ratio of lipids to water; the ratio level was set at 1 for lipids. The fat-to-water ratio was calculated in the vertebral bodies L1 and L5 and the intervertebral disc at L4/L5. A higher value of the fat/water ratio indicates a higher degree of degeneration.

### 2.3. Sample Preparation

Serum samples (200 µL) were deproteinized with 400 µL 1 mol/L HNO_3_; 1% Triton X-100 (200 µL) was added as a surfactant, mixed on a vortex and centrifuged for 10 min at 2000 rpm (MPW M-Diagnostic, Med. Instruments, Warszawa, Poland). Then, the supernatant was 15-fold diluted with 0.1 mol/L HNO_3_.

Blood samples (200 µL) were deproteinized with 800 µL 1 mol/L HNO_3_; 1% Triton X-100 (200 µL) was added as a surfactant, mixed on a vortex and centrifuged for 10 min at 2000 rpm (MPW M-Diagnostic, Med. Instruments, Warszawa, Poland). Then, the supernatant was 3-fold diluted with 0.1 mol/L HNO_3_.

### 2.4. Determination of Ca, Mg, Cd, Pb and Hg in Blood

The concentrations of mineral components in the serum (Ca, Mg) and in the whole blood (Cd, Pb) were determined by atomic absorption spectrometry (ASA) with acetylene-air flame atomization (Ca, Mg), electrothermal atomization (Cd, Pb) and Zeeman background correction using a Z-2000 instrument, Hitachi (Chiyoda, Japan). During the determination of Ca and Mg, a phosphate-masking reagent, −1% LaCl_3_, was added. During the determination of Cd and Pb, 0.5% NH_4_H_2_PO_4_ was added as a matrix modifier.

The Hg concentration in the whole blood was determined directly by the ASA method using the amalgamation technique (AMA-254, Leco, Pilsen, Czech Republic). Using this method does not require preliminary sample preparation. The blood samples were placed in a cuvette in the analyzer chamber, dried and then burned at 600 °C in an oxygen atmosphere, the Hg vapors being trapped by a gold amalgamator. The released Hg was measured in the following conditions: drying time—75 s, decomposition time—140 s, waiting time—45 s.

The control of the accuracy of the methods used was performed on certified reference materials of human serum and whole blood (Seronorm Trace Elements Serum L-1, 1309438; Seronorm Trace Elements Whole Blood L-2, 1702825; Sero-As, Hvalstad, Norway).

### 2.5. Statistical Analysis

The results obtained were statistically processed using the computer program Statistica v. 13 (Tibco, Palo-Alto, CA, USA). The data were tested for normal distribution by the Kolmogorov–Smirnov and Lilliefors tests. If the distribution showed asymmetry, non-parametric tests were used in further analyses. Similarly, when the data distribution was normal, parametric tests were used. Differences between groups were evaluated by Student’s *t*-test for data with normal distribution, and for data with a non-parametric distribution, the Mann–Whitney U-test was used to compare the differences between two groups, or the Kruskal–Wallis test was used to compare the differences between three groups. Correlation coefficients were assessed by Pearson’s test for normal data distribution or Spearman’s rank test for non-parametric data. The level of significance was assumed as *p* < 0.05.

## 3. Results

The serum Ca and Mg concentrations of subjects with osteoarthritis of the spine and those of the control group were compared. The concentration of Ca in the serum of patients with osteoarthritis of the spine did not significantly differ compared to the control group (Appendix A). The median serum Mg concentration of subjects with osteoarthritis of the spine (20.04 mg/L) was significantly lower (*p* < 0.001) than the median serum Mg concentration of healthy subjects (22.2 mg/L). The results are presented in Table 2.

The blood concentrations of Pb and Hg in the group of patients with osteoarthritis of the spine and those of healthy subjects did not differ significantly. No significant differences in Pb and Hg concentrations were found between men and women with osteoarthritis of the spine and the healthy population (Appendix A). In the control group, a significant correlation (Appendix A) was observed between blood Pb concentration and BMI (r = 0.42, *p* < 0.01), as well as between blood Pb concentration and age (r = 0.33, *p* < 0.01). In the healthy people, a significant correlation was also observed between age and Hg concentration in the blood (r = 0.56, *p* < 0.05). The median concentration of Cd in the blood of subjects with osteoarthritis of the spine (1.5 µg/L) was significantly higher (*p* < 0.05) than that in the control group (1.32 µg/L). The study group also had significantly higher (*p* < 0.001) median Cd concentrations among cigarette smokers than among non-smokers (Table 3).

The molar ratios of the concentrations of elements such as Ca (Table 4) and Mg (Table 5) to Pb, Cd and Hg were compared among subjects with osteoarthritis of the spine and the control ones. A significantly lower (*p* < 0.05) median molar ratio of Ca to Cd and Pb, as well as of Mg to Pb and Cd, was observed in the group of subjects with osteoarthritis of the spine compared to the control group.

There were no significant differences between the concentrations of the examined macronutrients and toxic elements and the pain intensity, assessed using a VAS scale.

In the conducted study, no significant correlations were found between the values of the fat/water ratio of the L1 and L5 vertebral bodies and the L4/L5 intervertebral disc obtained by ^1^H-MRS and the age, sex, BMI and pain level assessed by VAS (Appendix A). On the other hand, when the age groups were divided into under- and over-60-years groups (Table 6), a statistically significant difference was found between the two groups (*p* < 0.05) at the level of the L5 vertebral body. Among people over 60, the median fat/water ratio was higher.

An effect of cigarette smoking on the level of the fat/water ratio was observed (Table 7). The median fat/water ratios obtained by ^1^H-MRS at the L1 and L4/L5 intervertebral disc level were significantly higher in the smoking group (*p* < 0.05 and *p* < 0.001, respectively).

The levels of fat/water ratios in the L1 and L5 vertebral bodies and intervertebral disc at the L4/L5 level, obtained by ^1^H-MRS, were compared to the concentrations of the elements in relation to their blood or serum concentrations. The concentrations of Ca and Mg were assessed in three ranges: concentration values below normal, in the normal range and above normal. Toxic elements were divided into two categories: within the range of the acceptable standard and above the acceptable standard of element concentration in the blood. The analysis showed significant differences (*p* < 0.05) in the value of the fat/water ratio in the L5 vertebral body between patients whose serum Ca levels were below the recommended standard and those whose Ca levels were normal. The lowest fat/water ratio was observed in patients with a normal serum Ca concentration. (Figure 1).

The analysis showed a significant difference (*p* < 0.05) between the fat/water ratio of the L1 vertebral body and the L4/L5 intervertebral disc in terms of Mg concentrations below the recommended range versus concentrations of this element above the range (Figure 2). Higher fat/water ratio values were observed with the increased Mg concentration in the serum.

There were no statistically significant differences in the fat/water ratios obtained by ^1^H-MRS in the individual spinal sections in relation to Hg and Cd concentrations in the blood of patients. A significant low correlation (r = 0.214, *p* < 0.05) was observed only between the blood Pb concentration and fat/water ratio in the intervertebral disc at the L4/L5 level in the study group (Table 6).

## 4. Discussion

In the present study, the concentrations of selected elements in the blood of people with degenerative spine disease were evaluated. The results were related to the control group. The concentration of Ca in the two groups did not differ significantly. The literature data on Ca levels in osteoarthritis of the spine are inconsistent. In the studies of Zhao et al., the serum Ca content was significantly correlated with the degree of disk degeneration, which can suggest that the serum Ca concentration can be used as an indicator of intervertebral disk degeneration prognosis [20]. Khodyrev et al., in a study evaluating the serum levels of selected macronutrients and vitamins in patients with knee osteoarthrosis, showed Ca levels below the lower limit of the normal range [21]. In contrast, in a study conducted by Hunter et al. on bone turnover disorders and vitamin D_3_ and Ca regulation in osteoarthritis, no abnormalities were observed for the blood levels of Ca, Mg and phosphorus [22]. Various external factors, such as diet, physical activity, smoking cigarettes or taking medications, affect the serum Ca level [23]. In our study, we found no significant effect of cigarette smoking on serum Ca levels in both the test and control groups, which was probably related to the low number of cigarettes smoked during the day. Dongbo et al. showed that the serum Ca levels decreased significantly in the Chinese compulsive smokers and were related to the average number of cigarettes smoked per day and the degree of nicotine dependence [24]. In a study by Krall and Dawson-Hughes, smokers of at least 20 cigarettes per day had the lowest average fraction of Ca absorption. The results of the study also showed that smoking accelerates bone loss of the femoral neck and the whole body in older people [25]. In our study, we found that the median serum Mg concentration of people with osteoarthritis of the spine was significantly lower than the median serum Mg concentration of healthy people. Mg is an important element in maintaining proper homeostasis of the skeletal system. In the course of inflammatory and degenerative changes, there may be an increased loss of Mg in the body. Studies have shown that low serum Mg levels induced fibroblast apoptosis, and Mg supplementation alleviated chondrocyte apoptosis and facilitated chondrocyte proliferation and differentiation [26]. A study by Turgut et al. showed a close relationship between serum Mg concentration and intervertebral disc degeneration [27]. Different from the results in our study, those obtained Musik et al. showed that the Plasma Mg concentrations in patients with osteoarthritis of the spine and those in control subjects were not significantly different. In contrast, there was a significant difference between the concentrations in women in both groups, which may suggest a link between the higher prevalence of the disease in women and Mg deficiency in the body [28]. However, in our study, there was no difference in serum Mg concentration depending on sex. Another study observed a possible link between serum Mg levels and radiographic knee osteoarthritis. Serum Mg levels may have an inverse relationship with the radiological manifestation of knee osteoarthritis [29]. Serum Mg concentration was inversely related to the incidence of metabolic syndrome, diabetes, hypertension and hyperuricemia in patients with radiographic knee osteoarthritis [30].

The overexposure to toxic elements can cause a number of negative effects in the body. In our study, among those with osteoarthritis of the spine, 3% had blood Pb concentrations above acceptable standards. No blood Pb concentrations above the maximum allowable level were found in the control group. There were no significant differences in the concentration of Pb in the blood between the group of patients with osteoarthritis of the spine and healthy people. In our study, we also did not find significant differences in blood Pb concentration depending on sex. Potential differences in Pb concentration between men and women may result from different types of work and lifestyle, including dietary habits [31]. A study by Nelson et al. established links between blood Pb and biomarkers of joint tissue metabolism in osteoarthritis. This allowed us to demonstrate a negative effect of Pb on bone turnover and cartilage calcification in women and an effect on cartilage metabolism in men [32]. Among control subjects, a significant correlation (*p* < 0.01) was observed between Pb concentration and BMI. As BMI increases, blood Pb concentration increases (r = 0.42). The evaluation of blood Pb levels in a study conducted by Park et al. on a group of postmenopausal Korean women diagnosed with osteoarthritis, including lumbar spine joints, showed a significant association between osteoarthritis, BLL (Blood Lead Level) and BMI [33]. A significant correlation (*p* < 0.01) was observed between blood Pb levels and the age of subjects in the control group; Pb levels increased with age (r = 0.33). Pb is an element that accumulates in the body throughout life. Due to the limited possibilities of its removal and the slowness of the process, the half-life of Pb in the body is approximately 20 years [34]. Similar findings were obtained by Garrido Latorre et al. in a study conducted in Mexico on a group of women with osteoporosis [35]. We observed a significantly higher concentration of Cd in the blood of people with osteoarthritis of the spine compared to healthy people. Among those in the study group, 16% of people had blood Cd concentrations above the acceptable standard. In the control group, 5% of people had Cd concentrations above the maximum permissible value. The relationship between osteoarthritis of the spine and Cd exposure has not yet been studied in detail and extensively. There are still few data on the subject of individual elements and their correlation in cartilage and bone in patients with osteoarthritis degeneration. A recent study by Martinez-Nava et al. showed that Cd plays an antagonistic role with respect to the presence of essential elements such as zinc, iron and manganese. These micronutrients maintain cartilage homeostasis. Cd interferes with the absorption of these elements into cartilage, which promotes the development of diseases such as osteoarthritis and osteoporosis [36]. Jing et al. showed that Cd induces the apoptosis of fiber ring cells, which contributed to intervertebral disc degeneration through a mitochondria-related pathway [37]. A number of studies show a significant relationship between the concentrations of Cd in the blood and smoking cigarettes [38,39]. In our own study, in a group of people with osteoarthritis of the spine, significantly higher median blood Cd concentrations were found among smokers relative to non-smokers. We did not observe any significant differences in the Hg concentration in the blood between people with a degenerative disease of the spine and healthy people. It is known that exposure to mercury compounds adversely affects the functioning of the nervous system, including the skeletal system. Pamphlett et al. showed that Hg is taken up selectively, mainly by cells affected by rheumatoid arthritis and osteoarthritis of the spine. In addition, Hg is taken up by fibroblasts in several organs, often involved in multi-system connective tissue disorders. Hg induces autoimmune, inflammatory, genetic and epigenetic changes that have been described in a number of arthropathies and bone and connective tissue diseases [40]. In our study, both in the group of people with a degenerative disease of the spine and in the control group, there were no people whose Hg concentration exceeded the maximum permissible levels. The main source of human exposure to Hg is environmental pollution from heavy industry. Other sources include the frequent consumption of fish, certain medications, disinfectants, cosmetics and vaccines. Low Hg concentrations in the blood of our patients may result from the fact that they live in a clean area without heavy industry. For this reason, elevated Hg levels cannot be considered in our study as a predictive factor in osteoarthritis of the spine [41]. There was a significant positive correlation between Hg concentration and the age of patients in the control group. This may be due to the reduced Hg excretion capacity of the elderly [41].

In our own study, the molar ratio of the concentration of elements such as Ca and Mg to Pb, Cd and Hg was compared in the group of people with degenerative spine disease and in the control group. Significantly lower median ratios of Ca to Pb and Cd and Mg to Pb and Cd were observed among people with osteoarthritis of the spine compared to the control group. On this basis, we can propose a thesis that disturbed proportions between the studied macroelements and toxic elements, such as Pb and Cd, may predispose one to the occurrence of degenerative spine disease. Toxic elements can interact adversely with beneficial minerals and cause, among other things, disturbances in bone mineral homeostasis [42].

The concentrations of selected macronutrients and toxic elements were compared with VAS scores. We did not observe any significant differences in the degree of pain intensity depending on the concentration of Ca and Mg in the serum. Other results were obtained by Golan et al. in a study evaluating the correlations of pain intensity and biochemical parameters, especially Ca homeostasis. In this study, Ca was an independent risk factor for chronic pain. The resulting difference in mean serum Ca concentrations between the groups with and without pain was relatively small. In addition, Ca levels were correlated with pain severity, and in the logistic regression model, higher serum Ca concentrations were independently associated with chronic pain. The results supported the thesis that the relatively small difference in mean serum Ca levels is clinically relevant [43]. Adamczyk et al. described a type of spontaneous breakthrough pain caused by hypocalcemia or hypomagnesemia. This is a type of pain that usually does not respond to conventional analgesics until electrolyte disturbances are corrected [44]. Yousef et al. proved the effectiveness of intravenous Mg infusion and continued treatment with an oral Mg preparation among patients with chronic lumbar spine pain with a neuropathic component [45]. In our study, we did not find significant differences in blood Pb, Cd and Hg levels depending on the severity of pain. La-Up et al. found a positive association between urinary Cd concentrations and chronic musculoskeletal pain, suggesting that Cd exposure may cause neurological damage or damage to musculoskeletal tissue, resulting in chronic pain. In addition, Cd induces oxidative stress, which is associated with chronic muscular pain [46]. Ibrahim et al. also confirmed a positive correlation between the increased incidence of musculoskeletal complaints, bone pain, joint pain and muscle spasms and Cd levels in the blood and urine among Cd-exposed workers compared to a control group [47]. Among our patients, we did not have any people with occupational exposure to the effects of Cd compounds.

In our study, no significant correlations were found between the fat/water ratios of the L1 and L5 vertebral bodies and the L4/L5 intervertebral disc obtained by ^1^H-MRS and sex, age, BMI and VAS-assessed pain levels. A number of researchers have attempted to evaluate the fat and water contents in the vertebral body and intervertebral disc by ^1^H-MRS in order to assess the severity of osteoarthritis of the spine. A study by Kugel et al. showed that the spinal bone marrow fat content in women is lower than that in men in the same group and depends on age. In the age group of 31 to 50 years, the difference is greatest at 12%; in the youngest and oldest age groups, the difference is small [48]. Shellinger et al. evaluated the second lumbar vertebra using ^1^H-MRS with a single voxel technique and found an age- and sex-dependent linear increase in fat content within the vertebral bodies. Higher fat contents were found in men [49]. Griffith et al. showed that marrow fat content increases rapidly in women between the ages of 55 and 65, while, for men, marrow fat content increases gradually at a constant rate. Women over the age of 60 have a higher marrow fat content than men. This increased accumulation of marrow fat is consistent with the distribution of extra-bone fat in postmenopausal women [50]. Chen et al. demonstrated a sex-specific interaction between fat and bone density in older people. Higher marrow fat is correlated with lower trabecular bone mineral density in older women, but not in men [51]. In our study, after dividing into age groups (under and over 60), a statistically significant difference (*p* < 0.05) was found the in fat/water ratio at the L5 vertebral body. A higher fat/water ratio was found in older patients. We also observed an effect of cigarette smoking on the fat/water ratio. The median fat/water ratio obtained by ^1^H-MRS at the L1 vertebral body and L4/L5 intervertebral disc level was significantly higher in the smoking group (*p* = 0.019 and *p* = 0.001, respectively). In a study conducted by Nasto et al., exposure to tobacco smoke in an animal model study led to degenerative changes in the intervertebral disc and mineralization of the vertebral bodies. An increase in vertebral porosity and a decrease in the volume of the trabecular bone were observed. Intervertebral discs of smoke-exposed animals also showed a decrease in proteoglycan content and a decrease in prostaglandin synthesis [52]. The negative effect of cigarette smoking on intervertebral disc degenerative changes on MR imaging was also confirmed by Dogan et al. [53]. The relationship between bone marrow fat content and skeletal health has attracted widespread interest among researchers. MRI diagnosis and evaluation by ^1^H-MRS is now a standard technique for assessing bone marrow fat content. Marrow fat content has also been shown to be related to bone mineral density. One study reported a negative association between bone marrow fat content and osteoporosis. There is a suspected link between bone marrow fat and bone fracture in particular vertebral bodies [54].

In our study, the fat/water ratio levels at the L1 and L5 vertebral bodies and L4/L5 intervertebral disc were compared with the levels of examined mineral components. To date, no such study has been conducted. We observed significant differences (*p* < 0.05) in the value of the fat/water ratio in the L5 vertebral body between patients whose serum Ca levels were below the recommended standard and those whose Ca levels were in the reference range. The lowest fat/water ratio was observed in patients with a normal serum Ca concentration. The analysis showed a statistically significant difference in the fat/water ratio at the level of the L1 vertebral body and the L4/L5 intervertebral disc in terms of Mg concentrations below the norms compared to concentrations of this element above the recommended ranges. For the proper functioning of the body, not only is the right amount important, but the correct proportions between minerals are also important. Excessive amounts of one component may interfere with the bioavailability of another; in this case, excess Mg may impair Ca absorption. A tight control of Mg and Ca homeostasis seems to be crucial for bone health. On the basis of studies, both low and high Mg contents have harmful effects on the bones. Mg deficiency contributes to osteoporosis directly by acting on crystal formation and on bone cells and indirectly by impacting the secretion and the activity of the parathyroid hormone and by promoting low-grade inflammation. Less is known about the mechanisms responsible for the mineralization defects observed when Mg is elevated. Generally, controlling and maintaining Mg and Ca homeostasis is important to maintaining bone integrity [55,56].

There were also no statistically significant differences in the fat/water ratio in the different spinal segments in relation to Hg and Cd concentrations. A significantly positive low correlation was observed only between the blood Pb concentration and the fat/water ratio at the L4/L5 intervertebral disc, which may be associated with the intensification of degenerative changes in the event of exposure to Pb. Pb deposition has been observed in cartilage and bone in human osteoarthritis [57,58]. Prolonged exposure to Pb may affect bones and other joint structures. Pb disturbs the regulatory functions of bone cells and matrix synthesis by influencing the absorption and metabolism of Ca and the conversion of vitamin D [59,60,61]. Exposure to Pb influences the function of bone remodeling cells, causing impaired collagen synthesis by osteoblasts and the impaired resorptive capacity of osteoclasts [61].

Our study has several limitations that should be mentioned. We had no information on the diet or frequency of alcohol consumption in the study and control groups. We do not know if the eating habits did not differ in both groups, which could have influenced the obtained results. The lack of information about the medications used, the type of work and the level of physical activity, as well as the relatively small group of respondents, made it impossible to take into account the above factors in the study.

## 5. Conclusions

The results obtained may indicate the important role of disturbances between the concentration of the studied macronutrients and the concentration of toxic elements in the body in the course of osteoarthritis of the spine. The conducted research may help to approximate the complex etiology of such a common condition as osteoarthritis of the spine.

## Figures and Tables

**Figure 1 ijerph-19-11377-f001:**
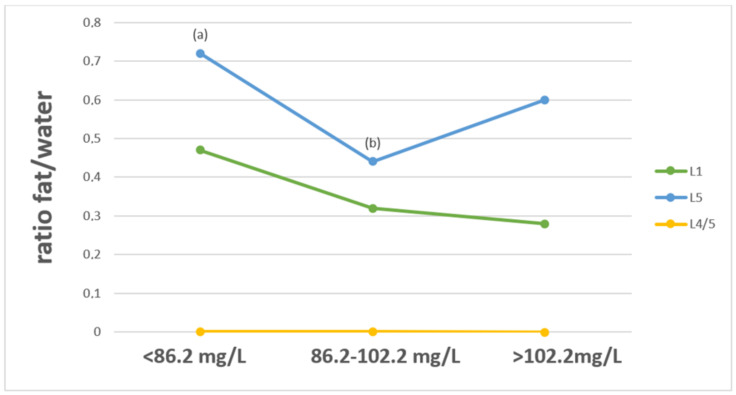
Fat/water ratio of selected spine structures in ^1^H-MRS, divided into serum Ca concentrations below, within and above the norm. Significant differences (*p* < 0.05) for (**a**) vs. (**b**) in the L5 vertebral body.

**Figure 2 ijerph-19-11377-f002:**
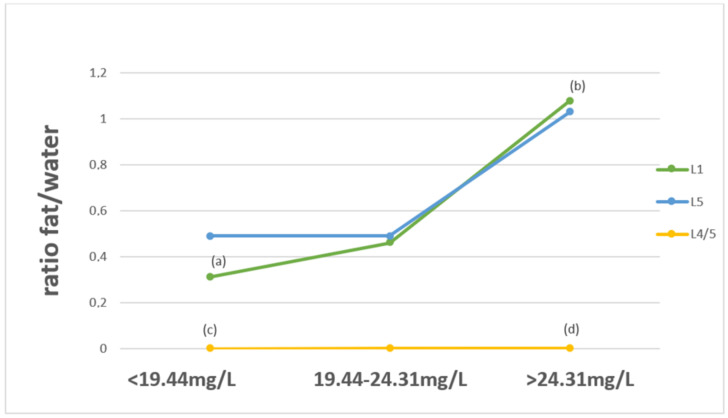
Fat/water ratio of selected spine structures in ^1^H-MRS, divided into serum Mg concentrations below, within and above the norm. Significant differences (*p* < 0.05) for (**a**) vs. (**b**) in the L1 vertebral body and for (**c**) vs. (**d**) in the L4/L5 intervertebral disc.

**Table 1 ijerph-19-11377-t001:** Characteristics of patients and the control group.

	Study Group *n* = 90		Control Group *n* = 40	
Age (mean ± SD) (range of years)	(48.3 ± 12.9) (21–73)		(45.8 ± 16) (21–83)	
Sex	Female 58%	Male 42%	Female 64%	Male 36%
BMI (kg/m^2^)	(28.1 ± 4.1) (20.1–38.3)		26.2 ± 3.8 (20.7–35.2)	
Smoking *	Yes 31%	No 69%	Yes 27%	No 73%

*n*—number of participants; SD—standard deviation; BMI—Body Mass Index; *—declared number of cigarettes smoked: 5–15 a day.

**Table 2 ijerph-19-11377-t002:** The concentration of magnesium in the serum of patients with osteoarthritis of the spine and healthy people.

Mg (mg/L); RV: 19.44–24.31 mg/L				
Group	Study Group		Control Group	
Sex	F	M	F	M
Mean ± SD	19.67 ± 3.00	19.66 ± 2.57	21.69 ± 2.52	22.20 ± 3.64
(min–max)	(11.42–26.56)	(15.40–25.42)	(16.90–25.62)	(15.41–27.66)
Median	20.06	19.62	22.08	22.20
(Q_1_–Q_3_)	(17.56–21.52)	(17.36–21.13)	(19.78–23.38)	(19.75–24.83)
Smoking	Smokers	Non-smokers	Smokers	Non-smokers
Mean ± SD	20.29 ± 2.52	20.12 ± 2.77	21.98 ± 2.55	22.16 ± 3.21
(min–max)	(15.40–25.42)	(14.60–26.56)	(18.72–25.73)	(15.41–23.66)
Median	20.5	20.16	21.6	22.85
(Q_1_–Q_3_)	(19.04–21.42)	(18.44–22.02)	(19.25–24.13)	(19.89–24.53)
All persons (total)				
Mean ± SD	19.67 ± 2.82		21.81 ± 2.88	
(min–max)	(11.42–26.56)		(15.41–27.66)	
Median	20.04 (A) **		22.2 (B)	
(Q_1_–Q_3_)	(17.48–21.34)		(19.78–24.13)	

RV—reference value; F—female; M—male; SD—standard deviation; min—minimum; max—maximum; Q_1_—lower quartile; Q_3_—upper quartile; **—statistically significant differences (*p* < 0.001) between the medians of the study group (A) and the control group (B) in the Mann–Whitney U-test.

**Table 3 ijerph-19-11377-t003:** The concentration of cadmium in the blood of patients with osteoarthritis of the spine and healthy people.

Cd (µg/L); RV < 2.7 µg/L				
Group	Study Group		Control Group	
Sex	F	M	F	M
Mean ± SD	1.9 ± 1.12	1.76 ± 1.02	1.38 ± 0.59	1.41 ± 0.63
(min–max)	(0.72–4.97)	(0.7–4.38)	(0.56–2.79)	(0.73–2.98)
Median	1.53	1.43	1.32	1.33
(Q_1_–Q_3_)	(0.97–2.51)	(0.82–2.56)	(0.96–1.56)	(0.86–1.59)
Smoking	Smokers	Non-smokers	Smokers	Non-smokers
Mean ± SD	2.20 ± 1.24	1.27 ± 0.59	1.40 ± 0.75	1.31 ± 0.48
(min–max)	(0.72–4.97)	(0.70–2.68)	(0.73–2.98)	(0.63–2.48)
Median	1.78 (A) **	1.08 (B)	1.25	1.3
(Q_1_–Q_3_)	(1.11–3.08)	(0.81–1.63)	(0.84–1.55)	(0.96–1.54)
All persons (total)				
Mean ± SD	1.84 ± 1.08		1.39 ± 0.59	
(min–max)	(0.70–4.97)		(0.56–2.98)	
Median	1.5 (C) *		1.32 (D)	
(Q_1_–Q_3_)	(0.93–2.55)		(0.94–1.57)	

RV—reference value; F—female; M—male; SD—standard deviation; min—minimum; max—maximum; Q_1_—lower quartile; Q_3_—upper quartile; **—statistically significant differences (*p* < 0.001) between the medians of smokers (A) and non-smokers (B) of the study group in the Mann–Whitney U-test; *—statistically significant differences (*p* < 0.05) between the medians of the study group (C) and the control group (D) in the Mann–Whitney U-test.

**Table 4 ijerph-19-11377-t004:** Molar ratios of Ca to Pb, Cd and Hg concentrations in patients with osteoarthritis of the spine and healthy people.

Group	Study Group	Control Group
Ca/Pb		
Mean ± SD	12,941 ± 4868	20,530 ± 16,014
(min–max)	(1082–28,313)	(5593–75,709)
Median	12,343 (A) *	13,408 (B)
(Q_1_–Q_3_)	(10,542–14,983)	(10,069–25,378)
Ca/Cd		
Mean ± SD	187,556 ± 90,600	234,852 ± 92,303
(min–max)	(43,709–366,799)	(101,899–472,590)
Median	177,714 (C) *	213,852 (D)
(Q_1_–Q_3_)	(110,809–268,423)	(178,525–273,468)
Ca/Hg		
Mean ± SD	613,280 ± 402,875	435,920 ± 183,293
(min–max)	(87,130–2,215,559)	(158,904–1,023,856)
Median	483,154	382,072
(Q_1_–Q_3_)	(322,554–838,270)	(312,726–557,396)

SD—standard deviation; min—minimum; max—maximum; Q_1_—lower quartile; Q_3_—upper quartile; *—statistically significant differences (*p* < 0.05) between the medians of the study group and the control group (A vs. B and C vs. D) in the Mann–Whitney U-test.

**Table 5 ijerph-19-11377-t005:** Molar ratios of Mg to Pb, Cd and Hg concentrations in patients with osteoarthritis of the spine and healthy people.

Group	Study Group	Control Group
Mg/Pb		
Mean ± SD	4657 ± 1871	7657.60 ± 5848.0
(min–max)	(372–10,320)	(2661.40–27,813)
Median	4552 (A) *	4993 (B)
(Q_1_–Q_3_)	(3129–5738)	(3941–9868)
Mg/Cd		
Mean ± SD	6821 ± 38,906	87,146.98 ± 36,315.8
(min–max)	(482–1418)	(30,449.10–184,765)
Median	62,238 (C) *	816,502 (D)
(Q_1_–Q_3_)	(32,584–101,698)	(60,345–112,976)
Mg/Hg		
Mean ± SD	22,705 ± 150,043	171,770 ± 72,321
(min–max)	(24,290–791,012)	(63,202–376,166)
Median	182,641	163,771
(Q_1_–Q_3_)	(113,744–305,413)	(112,431–221,853)

SD—standard deviation; min—minimum; max—maximum; Q_1_—lower quartile; Q_3_—upper quartile; *—statistically significant differences (*p* < 0.05) between the medians of the study group and the control group (A vs. B and C vs. D) in the Mann–Whitney U-test.

**Table 6 ijerph-19-11377-t006:** Fat/water ratio of selected structures of the spine in ^1^H-MRS in people with osteoarthritis of the spine before and after 60 years of age.

Group	Before 60 Years	After 60 Years
Ratio L1		
Mean ± SD	0.743 ± 0.176	0.753 ± 0.594
(min–max)	(0.012–0.840)	(0.152–1.891)
Median	0.307	0.624
(Q_1_–Q_3_)	(0.190–0.899)	(0.308–0.076)
Ratio L5		
Mean ± SD	3.652 ± 17.67	1.108 ± 0.493
(min–max)	(0.069–103.61)	(0.655–2.112)
Median	0.417 (A) *	0.883 (B)
(Q_1_–Q_3_)	(0.287–1.039)	(0.809–1.303)
Ratio L4/5		
Mean ± SD	0.0030 ± 0.0123	0.006 ± 0.0013
(min–max)	(0.000004–0.075)	(0.00003–0.036)
Median	0.001	0.001
(Q1–Q3)	(0.00005–0.0015)	(0.000043–0.0031)

SD—standard deviation; min—minimum; max—maximum; Q_1_—lower quartile; Q_3_—upper quartile; *—statistically significant differences in the L5 ratio (*p* < 0.05) between the medians of the study group before and after 60 years of age (A vs. B) in the Mann–Whitney U-test.

**Table 7 ijerph-19-11377-t007:** Fat/water ratio of selected spine structures in ^1^H-MRS in people with osteoarthritis of the spine disease depending on cigarette smoking.

Group	Smokers	Non-Smokers
Ratio L1		
Mean ± SD	1.032 ± 1.663	0.575 ± 0.516
(min–max)	(0.012–6.840)	(0.094–1.84)
Median	0.573 (A) *	0.303 (B)
(Q_1_–Q_3_)	(0.242–0.974)	(0.173–0.872)
Ratio L5		
Mean ± SD	0.791 ± 0.618	4.481 ± 19.818
(min–max)	(0.124–2.251)	(0.068–103.61)
Median	0.540	0.485
(Q_1_–Q_3_)	(0.330–1.276)	(0.274–1.061)
Ratio L4/5		
Mean ± SD	0.0037 ±.0093	0.0034 ± 0.014
(min–max)	(0.000025–0.036)	(0.000004–0.0748)
Median	0.001 (C) **	0.0001(D)
(Q1–Q3)	(0.000056–0.0026)	(0.000049–0.0014)

SD—standard deviation; min—minimum; max—maximum; Q1—lower quartile; Q3—upper quartile; *—statistically significant differences in the L1 ratio (*p* < 0.05) between the medians of smokers (A) and non-smokers (B) of the study group in the Mann–Whitney U-test; **—statistically significant differences in the L4/L5 ratio (*p* < 0.001) between the medians of smokers (C) and non-smokers (D) of the study group in the Mann–Whitney U-test.

## Data Availability

The data presented in this study are available on request from the corresponding author. The data are not publicly available due to privacy concerns.

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
