# Peer review of "Concentration of Selected Macronutrients and Toxic Elements in the Blood in Relation to Pain Severity and Hydrogen Magnetic Resonance Spectroscopy in People with Osteoarthritis of the Spine"

_ijerph, 2022, doi:10.3390/ijerph191811377_

Round 1
Reviewer 1 Report
Reviewer comments
General Comments
· Language must be improved
· Grammar must be improved
· Revision needed to relay clear concepts
Abstract
· The abstract is too long. Instructions to authors says “The abstract should be a total of about 200 words maximum”
· Never start a sentence with a number
· The objective is unclear: define more clearly the relationship(s) you want to investigate.
· If you mention that something is significantly greater/smaller: you must put the error +
· No association or results presented for the VAS; relevant or not?
Introduction
· Missing references
· Highlighted parts needs revision either due to grammar, language, or sentence structure. Highlight in the pdf also indicate that the concept of the sentence is not adequate.
Needs...more information. Impact of nutrients, acceptable ranges etc
Methods
· For all equipment, model numbers are necessary.
· The methods for the analysis of the serum (sera??) and blood samples is unclear and needs to be re-written. Use smaller sentences if necessary.
· Statistics: normalcy checked, but no mention of Homogeneity of Variances
· “Differences between groups were 126 shown by post-hoc test”: specify
Results
· · define parameters (p value) and tests - for each table. The table footer should be a comprehensive synopsis
· Some results discussed are not significant; if they are please add p value in the text
· When discussing results add the table reference the first time it is mentioned
· Results discussing mineral levels needs work. Results mentioned are either not significant, or not (of the ones that are significant) they are clearly stated in the text with values and p value.
· Figure 1: very poor quality . This is from excel; they can be sized properly.
· Figure 1…where is the range (error?) same for figure 2
· Be consistent with terminology : osteoarthritis, or degenerative spine? Figure 1 and 2 have different terms.
· Data displayed or analysed by L1 and L5 vertebral bodies and the L4/L5 intervertebral disc…is missing or not presented? There is a whole discussion, but no results?
Discussion
· The discussion feels like a list of other people’s work with 1 line sentence to described their results and then 1 lines about the current results. There is no “discussion” about the links, trends, theory, expansion or explanations of the results, justifications, theories, etc.
· The section on Cd was better explained; but again poorly
· The discussion was one giant chunky paragraph; no sections to guide the reader and was very confusion to try to put together a “story” or impact of your work
· Discussion mentions results …which have no table or figure form the results section.
Author Response
Dear Reviewer,
Thank you very much for taking the time to review our manuscript and for all your comments. The following are the point-by-point replies to your comments.
General Comments
Language must be improved
Grammar must be improved
Thank you, sorry for our mistakes. We corrected English language. I would like to inform that linguistic correction will be also outsourced to MDPI.
Revision needed to relay clear concepts
Thank you, we tried to improve the manuscript as best we can.
Abstract
The abstract is too long. Instructions to authors says “The abstract should be a total of about 200 words maximum”
Thank you for this remark and we agree with it as much as possible. We reduced the abstract by about 100 words.
Never start a sentence with a number
Thank you. We corrected this sentences.
The objective is unclear: define more clearly the relationship(s) you want to investigate.
Thank you. You are absolutely right. We have clarified the objective of our research.
If you mention that something is significantly greater/smaller: you must put the error +
Thank you for this remark. In order to limit the number of words in the abstract, we have removed the numerical values. We have included the above remark in the results section.
No association or results presented for the VAS; relevant or not?
Thank you, we've added these results.
Introduction
Missing references
Thank you. We added references in the introduction
Highlighted parts needs revision either due to grammar, language, or sentence structure. Highlight in the pdf also indicate that the concept of the sentence is not adequate.
Thank you. We revised, but we couldn't find enclosed pdf version. linguistic correction will be also outsourced to MDPI.
Needs...more information. Impact of nutrients, acceptable ranges etc
Thank you. We improved introduction and added suggested informations.
Methods
For all equipment, model numbers are necessary.
We added missing model numbers of the equipment used in the experiments.
The methods for the analysis of the serum (sera??) and blood samples is unclear and needs to be rewritten. Use smaller sentences if necessary.
Thank you for this valuable attention. The methodology, including sample preparation, has been redrafted.
Statistics: normalcy checked, but no mention of Homogeneity of Variances
Differences between groups were 126 shown by post-hoc test”: specify
We added information about the normality distribution and homogeneity of variances tests and clarified what comparisons we used the post-hoc test for.
Results
define parameters (p value) and tests - for each table. The table footer should be a comprehensive synopsis
Thank you, we have completed this.
Some results discussed are not significant; if they are please add p value in the text
Thank you, we have added p values or indicated not significant results.
When discussing results add the table reference the first time it is mentioned
Thank you, we have improved it.
Results discussing mineral levels needs work. Results mentioned are either not significant, or not (of the ones that are significant) they are clearly stated in the text with values and p value.
Thank you. We edited this part of the results, added p values.
Figure 1: very poor quality . This is from excel; they can be sized properly.
Thank you. We tried to improve the quality of the figure.
Figure 1…where is the range (error?) same for figure 2
Thank you. We have added error bars in the figures.
Be consistent with terminology : osteoarthritis, or degenerative spine? Figure 1 and 2 have different terms.
Thank you, we apologize for our inconsistency. We have standardized the captions of the figures.
Data displayed or analysed by L1 and L5 vertebral bodies and the L4/L5 intervertebral disc…is missing or not presented? There is a whole discussion, but no results
Thank you for this valuable tip. We added additional tables and figures to the results.
Discussion
The discussion feels like a list of other people’s work with 1 line sentence to described their results and then 1 lines about the current results. There is no “discussion” about the links, trends, theory, expansion or explanations of the results, justifications, theories, etc.
Thank you for this valuable comment. We improved the discussion as best we can.
The section on Cd was better explained; but again poorly
Thank you. We corrected section concerned Cd.
The discussion was one giant chunky paragraph; no sections to guide the reader and was very confusion to try to put together a “story” or impact of your work
Thank you. We have organized the discussion as best we can and divided it into sections.
Discussion mentions results …which have no table or figure form the results section.
We improved it. We have added additional tables and figures in the results part
Thank you once again for all your comments and for taking your precious time to read and review our manuscript. Corrections made according to your comments will increase the value of our article.
Authors

Reviewer 2 Report
This is an interesting explorative study. However the manuscript needs to be better organized and written. It needs to be logically written coming with a rational for the study with objectives of the study clearly stated. It looks to me an explorative study.
The authors did a series of statistical tests. I am not qualified to assess the statistical methods. A qualified statistician should review this manuscript.
Limitations of the study need to be mentioned.
Author Response
Dear Reviewer,
Thank you very much for taking the time to review our manuscript and for all your comments. The following are the point-by-point replies to your comments.
This is an interesting explorative study. However the manuscript needs to be better organized and written. It needs to be logically written coming with a rational for the study with objectives of the study clearly stated. It looks to me an explorative study.
Thank you for your valuable comment. We have re-edited the manuscript as suggested. We tried to improve the manuscript as best we can.
The authors did a series of statistical tests. I am not qualified to assess the statistical methods. A qualified statistician should review this manuscript.
Thank you, we improved the description of statistical methods as suggested by the Reviewer 1
Limitations of the study need to be mentioned.
Thank you for your valuable comment. We added limitations of the study.
Thank you once again for all your comments and for taking your precious time to read and review our manuscript. Corrections made according to your comments will increase the value of our article.
Authors

Round 2
Reviewer 1 Report
General comments
Significant improvements have been completed on the manuscript. There are still concerns regarding statistics and results that need to be addressed.
Abstract
General formatting and language should be looked over in abstract and text.
Eg. Space missing : theosteoarthritis of the sp
Space missing : Ca andMg and Pb
Too many ands (use commas),, Ca andMg and Pb, Cd and Hg in blood with the
Materials and Methods
-Please make sure that you mean gender and not sex. If using gender, please describe criteria and how data was collected for gender.
You must define the 1H for the NMR: Hydrogen 1 (1H) magnetic resonance (MR) spectroscopy
Section 2.3. Sample preparation Is confusing and needs to be re-written
Information on age, gender and smoking status was collected……where is it though, this should be in a table. This is very pertinent as a large part of your analysis discusses smoker VS non smoker and sex…it is needed to know how many smoker VS non were there, etc ? This is all missing. This “The control group was appropriately matched in terms of gender, age and percentage of smokers and non-smokers, relative to the group with degenerative spine disease” is not enough
Statistics
The Kruskal-Wallis ANOVA test…..Kruskal-Wallis test is the non-parametric equivalent of an ANOVA (analysis of variance). Kruskal-Wallis is used when researchers are comparing three or more independent groups on a continuous outcome, but the assumption of homogeneity of variance between the groups is violated in the ANOVA analysis
In other words: the Kruskal–Wallis test (1952) is a nonparametric approach to the one-way ANOVA. The procedure is used to compare three or more groups on a dependent variable that is measured on at least an ordinal level
The KW test is NOT an ANOVA.
Line 160 :…” The Kruskal-Wallis ANOVA test and post-hoc test were used to show differences in” …..is therefore not correct.
A full review of the stats is needed, as a basic understanding of these tests are concerning.
If your methods say p, 0.05 is significant, then why are your reporting p<0.0005 line 178 ???
An interesting read: Weimo Zhu, p < 0.05, < 0.01, < 0.001, < 0.0001, < 0.00001, < 0.000001, or < 0.0000001 …, Journal of Sport and Health Science, Volume 5, Issue 1, 2016, Pages 77-79, ISSN 2095-2546,
The table footer description the stats is still not well done. Please refer to other articles and use the proper format
Results
The results…need to be critically assessed,,,,eg.why are you bothering to investigate that the median MG in OA spine is lower then MEAN serum in healthy? Why are you comparing two different metric among two different populations?
Language needs to be revised for the whole document . EG Line 187…sick people..is not a well defined term.
IF results are not significant, why point them out? If you are wanting to discuss it; then you cannot make an assumption that one population is greater then the other
Eg. You say: In both groups,higher Pb concentrations were found in men compared to women, but the differences were not statistically significant.
Should be: : no significant difference in Pb concentration were found between men and women in OA and health populations. (WE cannot say men > then women. BECAUSE its not significant)
For results that you don’t discuss (eg Hg for the most part)….put in supplemental.
You say “Subjects with serum Mg levels within the normal range had the least pain assessed on the 238
VAS scale, but the differences were not statistically significant. Both deficiency and excess 239
of Mg in the blood serum of people with osteoarthritis of the spine may be responsible for 240
the severity of pain.”…….Your results show that MG levels are not significant associated with pain…so your second sentence is false and misleading. You are stretching the results to fit a certain hypothesis, rather then critically looking at the data.
For tendencies….a trend needs a p value. Again stretching results to fit within a concept
Figures…need much improvement. Eg. The call out values do not follow significant figure rules, etc, no titles.
The data, results and discussion need to by in sync. A lot of your data show no relationship, which can deviate from the current literature. That is FINE. BUT Discuss this different and propose why
Author Response
Dear Reviewer,
Thank you once again for all your comments and for taking your precious time to reread and review our manuscript. Below are the point-by-point responses to your comments.
General comments
Significant improvements have been completed on the manuscript. There are still concerns regarding statistics and results that need to be addressed.
Thanks for your valuable comments and your time again. We made significant changes to the manuscript again. First of all, according your valuable suggestion, we have edited the results and the discussion in accordance with the results of the statistical analysis.
Abstract
General formatting and language should be looked over in abstract and text.
Eg. Space missing : theosteoarthritis of the sp
Space missing : Ca andMg and Pb
Too many ands (use commas),, Ca andMg and Pb, Cd and Hg in blood with the
We apologize for these errors, they have all been corrected. The lack of spaces between words was due to a poor version of Word where this mistakes were invisible. Moreover, linguistic correction will be outsourced to MDPI.
Materials and Methods
-Please make sure that you mean gender and not sex. If using gender, please describe criteria and how data was collected for gender.
Thank you for this comment. We swapped gender for sex. We qualified subjects for gender on the basis of data included in the questionnaires of patients and healthy people.
You must define the 1H for the NMR: Hydrogen 1 (1H) magnetic resonance (MR) spectroscopy
Thank you. We defined it. Proton spectroscopy is used interchangeably for hydrogen spectroscopy. Indeed, in some places in the manuscript we did not specify this type of method .
Section 2.3. Sample preparation Is confusing and needs to be re-written
Thank you. We improved section concerning preparation of sample. We also added more details on the direct determination of mercury in section 2.4.
Information on age, gender and smoking status was collected……where is it though, this should be in a table. This is very pertinent as a large part of your analysis discusses smoker VS non smoker and sex…it is needed to know how many smoker VS non were there, etc ? This is all missing. This “The control group was appropriately matched in terms of gender, age and percentage of smokers and non-smokers, relative to the group with degenerative spine disease” is not enough
Thank you for this valuable comment. We put the data on the characteristics of the patients and the control group in Table 1.
Statistics
The Kruskal-Wallis ANOVA test…..Kruskal-Wallis test is the non-parametric equivalent of an ANOVA (analysis of variance). Kruskal-Wallis is used when researchers are comparing three or more independent groups on a continuous outcome, but the assumption of homogeneity of variance between the groups is violated in the ANOVA analysis
In other words: the Kruskal–Wallis test (1952) is a nonparametric approach to the one-way ANOVA. The procedure is used to compare three or more groups on a dependent variable that is measured on at least an ordinal level
The KW test is NOT an ANOVA.
Line 160 :…” The Kruskal-Wallis ANOVA test and post-hoc test were used to show differences in” …..is therefore not correct.
We apologize for these inaccuracies. This test was used to show differences in the degree of spinal degeneration fat/water ratio and in the VAS scale in subjects with osteoarthritis of the spine according to normal or abnormal blood levels of the element in question - comapring three gropus: below, within the range and above the range of reference level.
In our computer program, Statistica, there is such a nomenclature, perhaps not entirely correct. Hence this ambiguity in the name of the test used.
Please see picture in attached file.
A full review of the stats is needed, as a basic understanding of these tests are concerning.
If your methods say p, 0.05 is significant, then why are your reporting p<0.0005 line 178 ???
An interesting read: Weimo Zhu, p < 0.05, < 0.01, < 0.001, < 0.0001, < 0.00001, < 0.000001, or < 0.0000001 …, Journal of Sport and Health Science, Volume 5, Issue 1, 2016, Pages 77-79, ISSN 2095-2546,
Thank you very much. We have revised the description of the statistics throughout the manuscript according to your suggestion.
We read an article by Weimo Zhu. These are valuable considerations for the truth of the statements based on statistical tests and levels of significance. Our statistical analyzes, however, were based on generally accepted principles, used in most scientific articles. P<0.05 means theoretically that 95% of the cases meet the assumption. Of course, it is true that significance depends on the size of the groups. We agree that the p-values we have reported have introduced a certain mess. Therefore, in line with your suggestion, we standardized the p values. However, to emphasize the significance of our results, in line with the general statistical principles, we adopted three levels: p <0.05, p <0.01, p <0.001.
The table footer description the stats is still not well done. Please refer to other articles and use the proper format
Thank you. We have corrected the footers in the tables in accordance with your suggestion and the pattern of other articles, so that there is clarity between which parameters are significant differences. However, we looked at recent articles published in IJERPH and found that there is no single style for describing footers in tables.
Results
The results…need to be critically assessed,,,,eg.why are you bothering to investigate that the median MG in OA spine is lower then MEAN serum in healthy? Why are you comparing two different metric among two different populations?
We apologize for this mistake. This is our unfortunate mistake in the word used. In both cases, we meant the median. In parentheses, we have given medians and quartiles both for people with degenerative spine disease and for the control group. We also mention medians in the discussion.
Language needs to be revised for the whole document . EG Line 187…sick people..is not a well defined term.
Thank you. We have removed this unfortunate wording.
IF results are not significant, why point them out? If you are wanting to discuss it; then you cannot make an assumption that one population is greater then the other
Thank you for this valuable suggestion. We improved the description of the results, especially paying attention not to suggest differences between groups if they were not statistically significant.
Eg. You say: In both groups,higher Pb concentrations were found in men compared to women, but the differences were not statistically significant.
Should be: : no significant difference in Pb concentration were found between men and women in OA and health populations. (WE cannot say men > then women. BECAUSE its not significant)
Your suggestion is absolutely right. We have changed the description so as not to suggest trends that are not significant.
For results that you don’t discuss (eg Hg for the most part)….put in supplemental.
As you suggested, we transferred the statistically insignificant results to supplementary file, in addition to the table with Hg, also results concerning Ca and Pb.
You say “Subjects with serum Mg levels within the normal range had the least pain assessed on the 238
VAS scale, but the differences were not statistically significant. Both deficiency and excess 239
of Mg in the blood serum of people with osteoarthritis of the spine may be responsible for 240
the severity of pain.”…….Your results show that MG levels are not significant associated with pain…so your second sentence is false and misleading. You are stretching the results to fit a certain hypothesis, rather then critically looking at the data.
You are absolutely right. We wanted to show some tendencies between the severity of perceived pain and the concentration of the assessed minerals. However, these differences were not statistically significant, so we removed this description and the figures on VAS versus element concentrations.
For tendencies….a trend needs a p value. Again stretching results to fit within a concept
Thank you for this comment. We have corrected the description of the results so as not to suggest a trend if the differences are not significant.
Figures…need much improvement. Eg. The call out values do not follow significant figure rules, etc, no titles.
Thanks for your comment. We improved the values on the graphs, added information about the significance in the caption of the figures.
The data, results and discussion need to by in sync. A lot of your data show no relationship, which can deviate from the current literature. That is FINE. BUT Discuss this different and propose why
Thank you for valuable comment. We tried to correct the discussion again as best we could, above all to synchronize it with the revised results to avoid bias in our descriptions.
Thank you again for all your comments, which significantly improved the quality of our manuscript.
Authors

Round 3
Reviewer 1 Report
The manuscript has greatly improved; the authors have done a very good job in responding to the feedback
· Abstract was significantly improved.
o Add p values when mentioning significance in the abstract
· Materials and methods section has greatly improved. No comments.
· Results
o If the MG median is significantly different (and you’ve specified the value and the table ) don’t bother mention the Q1 and Q3.
o The use of “examined group” can get a little confusing. Please change to diagnosed, patients or study group (as described in your methods section 2.1)
o Line 286-287 mentions correlation. The stats must be described in the stats section. (Pearson correlation?).
o Please add correlation results in a table – this is very interesting.
o Please note or add non significant corrections in supplemental- may be useful for future work
o Lines 336-339 the median of the molar ratios difference not the molar ration (which usually implies mean)
o The Q3 of Ca/pB have changed….please verify
o Footers have improved to help guide readers. Good
o Line 357 “assessed using a scale” not “on” a scale (minor edit)
o Results mentioned in lined 358-360 would be great to add in a table in supplemental.
o Figure is clear (although I still prefer to not have data call out points). Just the graph is enough (people can estimate the fat/water ratio value.
· The discussion has greatly improved.
By the way
· Your comment: Therefore, in line with your suggestion, we standardized the p values. However, to emphasize the significance of our results, in line with the general statistical principles, we adopted three levels: p <0.05, p <0.01, p <0.001.
o Re-answer: Yes. Just to let you know the paper was not fully directed at your case specifically. It is a helpful read generally.
o But the P values by the three levels is an excellent convention to indicate significance. Good.
Author Response
Dear Reviewer
Thank you again for your precious time, your patience and any valuable suggestions that have improved the quality of our publication.
The manuscript has greatly improved; the authors have done a very good job in responding to the feedback
Thank you for your nice opinion.
- Abstract was significantly improved.
Add p values when mentioning significance in the abstract
Thank you for this comment. We have added the p values in the abstract. We also deleted the mistakenly left sentence regarding the relationship between the severity of pain and the concentration of the elements, as these differences were insignificant. We also improved the sentence regarding the fat/water ratio in relation to serum Ca and Mg concentrations.
- Materials and methods section has greatly improved. No comments.
Thank you.
- Results
o If the MG median is significantly different (and you’ve specified the value and the table ) don’t bother mention the Q1 and Q3.
Thank you for your good comment, we've removed the quartiles from the text.
o The use of “examined group” can get a little confusing. Please change to diagnosed, patients or study group (as described in your methods section 2.1)
Thank you for suggestion. You're right, we've removed that confusing term.
o Line 286-287 mentions correlation. The stats must be described in the stats section. (Pearson correlation?).
Thank you for valuable comment. We described correlation tests in the Statistics section. Correlation coefficients were assessed by Pearson's test for normal data distribution or Spearman's rank test for non-parametric data.
o Please add correlation results in a table – this is very interesting.
Thank you for suggestion . We have added tables with correlations in the supplementary file.
o Please note or add non significant corrections in supplemental- may be useful for future work
Thank you for suggestion . We have added tables with correlations in the supplementary file.
o Lines 336-339 the median of the molar ratios difference not the molar ration (which usually implies mean)
You are right, our description was misleading. We added the word: median
o The Q3 of Ca/pB have changed….please verify
Thank you very much. Of course, this is our mistake when redrafting the table. We corrected it.
o Footers have improved to help guide readers. Good
Thank you for your opinion.
o Line 357 “assessed using a scale” not “on” a scale (minor edit)
You are right. Sorry for the mistake. We corrected it.
o Results mentioned in lined 358-360 would be great to add in a table in supplemental.
Thank you. We have added these results in the next table in the supplementary file.
o Figure is clear (although I still prefer to not have data call out points). Just the graph is enough (people can estimate the fat/water ratio value.
As per your suggestion, we have removed the numerical values from the charts, only the significance marks left.
- The discussion has greatly improved.
Thank you for your nice opinion.
By the way
- Your comment: Therefore, in line with your suggestion, we standardized the p values. However, to emphasize the significance of our results, in line with the general statistical principles, we adopted three levels: p <0.05, p <0.01, p <0.001.
Thank you.
o Re-answer: Yes. Just to let you know the paper was not fully directed at your case specifically. It is a helpful read generally.
Thank you. We understand and agree with you that the statistics are based on some general assumptions and this is some kind of estimation with assumed uncertainty.
o But the P values by the three levels is an excellent convention to indicate significance. Good.
Thank you, we agree that this brings some order and we will apply these principles in future publications.
Thank you again for a very careful analysis of our publication and all suggestions.
Authors
